CD81 inhibits the proliferation of acute megakaryoblastic leukemia cells

Su Narun 18120697060@163.com
Hu Xiaohao drhydra@163.com
Department of Pediatrics, The Second Affiliated Hospital of Fujian Medical University , Quanzhou , Fujian , China
Uversky Vladimir
Electronic publication date: 2025 Nov 5
Publication date: 2025
Volume: 13
Electronic Location ID: e20286
Received 2025 Mar 10; Accepted 2025 Oct 2
Copyright: ©2025 Su and Hu
Copyright year: 2025
Copyright holder: Su and Hu
License: This is an open access article distributed under the terms of the Creative Commons Attribution License, which permits unrestricted use, distribution, reproduction and adaptation in any medium and for any purpose provided that it is properly attributed. For attribution, the original author(s), title, publication source (PeerJ) and either DOI or URL of the article must be cited.
License URL: https://creativecommons.org/licenses/by/4.0/

Keywords: CD81, AMKL, Target, Leukemia

Funding: The authors received no funding for this work.

==============================
CD81 is a cell surface protein that plays an important part in tumor development. Several studies have shown that CD81 plays a role in cancer cell proliferation, invasion, and metastasis, particularly in leukemia. It has been reported that CD81 is overexpressed in non-Down syndrome acute megakaryoblastic leukemia (non-DS AMKL). In this article, we hypothesize that CD81 may play a vital role in acute megakaryoblastic leukemia (AMKL). We constructed the CD81 knockdown cell line using shRNA and found that CD81 knockout can inhibit the proliferation of AMKL and increase the apoptosis of AMKL in vitro. Therefore, CD81 may be a target of AMKL.

Introduction

Acute megakaryoblastic leukemia (AMKL) is a myeloid malignancy characterized by the excessive proliferation of abnormal megakaryocytes (MKs) (McNulty & Crispino, 2020). The World Health Organization (WHO) diagnostic criteria for AMKL include: the percentage of blasts in bone marrow should be ≥20% and more than half are derived from the megakaryocytic lineage; and the blasts express at least one platelet-specific antigen, such as CD41, CD42b, or CD61, which can be detected using flow cytometry (Swerdlow et al., 2017). AMKL can be classified into Down syndrome AMKL (DS-AMKL) and non-DS AMKL subtypes, and the latter is the focus of this article. Pediatric AMKL is rare and accounts for 4%–15% of newly-diagnosed AML cases (De Rooij et al., 2017; Luo, Yu & An, 2021). Non-DS AMKL is a heterogenous disease associated with a poor prognosis. The reported overall survival (OS) of non-DS-AMKL ranges from 13% to 70% (De Marchi, Araki & Komatsu, 2019; Hama et al., 2008; Qi et al., 2020). The current therapeutic options available for the treatment of AMKL are chemotherapy and hematopoietic stem cell transplantation (HSCT). Advancements in chemotherapy regimens and the usage of HSCT following chemotherapy have improved the remission rates of AMKL, with the 3-year OS reaching up to 82% in recent years (Hahn et al., 2016). However, this is still far from satisfactory, and the target therapies of AMKL are under active investigation (Li & Kalev-Zylinska, 2023). To date, targets such as NCAM1 (CD56) (Smith et al., 2020), FOLR1 (Tang et al., 2022), Aurora A, and BCL-XL (Gress et al., 2024) are promising prospects. In our previous study (Su et al., 2022), we also identified CD81 as another potential target for AMKL.

CD81 plays an important role in the immune system, viruses, and tumors (Bunz et al., 2024; Carloni, Mazzocca & Ravichandran, 2004; Hasezaki, Yoshima & Mine, 2020; Li et al., 2020). According to Na et al.’s (2024) work, the increased expression of CD81 was observed in T cells with increased cytotoxicity. CD81, CD9, and CD63 are components of extracellular vesicles (EVs) (Fan et al., 2023), which have an association with SARS-CoV-2 (Sun et al., 2023). CD81 plays a crucial role in cancer cell proliferation, invasion, and metastasis (Floren & Gillette, 2021; Kowalczyk et al., 2023; Titu et al., 2021). CD81 was implicated in B-cell and mantle cell lymphomas, acute lymphoblatic leukemia (ALL), and acute myeloid leukemia (AML) (Bailly & Thuru, 2023). However, research on CD81 in hematological malignancies specifically is still very limited. To date, it has only been found to be a marker of poor prognosis in AML. Therefore, our research on the role of CD81 in AMKL is innovative.

Considering the overexpression of CD81 in AMKL cells, we postulate that knockdown of CD81 may inhibit the proliferation of AMKL.

Materials & Methods

Cell culture

Human AMKL cell line UT-7 was purchased from Fuheng Cell Center, Shanghai, China. All cells were maintained in Roswell Park Memorial Institute 1,640’s medium (RPMI 1640, Fuheng Corporate, Shanghai, China) supplemented with 10% fetal bovine serum (FBS, Hyclone Laboratories, Inc., Logan, UT, USA) and 1% penicillin/streptomycin (Sangon Biotech Co., Ltd, Shanghai, China). Cells were incubated in a thermostatic incubator at 37 °C with 5% CO2.

Screening of the best target sequence

The siRNA targeting CD81 was synthesized by Gencefe Corporate, Jiangsu, China. The siRNA-1 (CD81-human-498), siRNA-2 (CD81-human-168), and siRNA-3 (CD81-human-636) that target distinct regions of the CD81 mRNA, which helps rule out off-target effects and sequences, are shown in Table S1. UT-7 cells were transfected with siRNAs using Lipo8000 (Beyotime, Shanghai, China) according to the manufacturer’s protocol. Real-time polymerase chain reaction (RT-PCR) analysis was conducted to evaluate the interfering efficiency. The sense sequence of the CD81 primer was 5′-TGACCCGCAGACCACCAAC-3′. The siRNA-2 sequence was chosen for further research.

Total RNA isolation and quantitative real-time polymerase chain reaction analysis

Total RNA from the AMKL cell line UT-7 was isolated using Trizol reagent (Invitrogen, Waltham, MA, USA) according to the manufacturer’s protocol. A Smartspec Plus (Bio-Rad, Hercules, CA, USA) was used to investigate the quality and concentration of RNA, and 5 µL total RNA (concentration: 25%) was used to synthesize cDNA using a 1st strand cDNA synthesis kit (Aidlab Biotechnologies Co., Ltd., Beijing, China). SYBR green reagent (Aidlab, China) was used to perform qPCR analysis on an ABI7500. Relative mRNA expression level was quantified using the 2−ΔΔCt method.

Construction of stable CD81 knockdown cell line

For stable knockdown of the CD81 gene, LV-shCD81 (a lentiviral vector encoding a short hairpin RNA (shRNA) specifically designed to knock down CD81 expression) lentiviral particles were purchased from Shanghai GenePharma, Shanghai, China. AMKL cells were infected with the indicated lentivirus. Stable cell lines were selected using puromycin (0.4 µg/mL, 72 h).

Cell proliferation assay

A Cell Counting Kit-8 (CCK8, CTCC, Jiangsu, China) was used to examine the proliferation ability of AMKL cells. Cells were seeded into a 96-well plate at a density of 5 × 103 cells per well and cultured for 48 h. We then added 10 µL CCK8 reagents into each well and incubated them for 2 h at 37 °C. The Smartspec Plus (Bio-Rad, Hercules, CA, USA) was used to measure the absorbance at the wavelength of 450 nm.

Detection of cell cycle

When the cells were cultured to the logarithmic stage, the cell concentration was adjusted to 2×105 cells/mL. Cells were seeded into a six-well plate at a density of one mL per well and cultured for 48 h. The cells were washed with phosphate-buffered saline (PBS) and 70% ethanol was added slowly to pre-washed cells while vortexing and fixed overnight at 4 °C. The cells were then washed with PBS again, RNaseA (100 µg/mL) was added, and they were incubated for 30 min at 37 °C. Propidium iodide (PI, 50 µg/mL) was added into the cells and incubated at 37 °C for 30 min without light. The cells were washed with PBS and the cell cycle was detected using a flow cytometer.

Figure 1 Screening of the best target sequence.

Expression of CD81 mRNA in UT-7 cells was detected by RT-PCR. The horizontal axis indicates different group of UT-7 cells, and the vertical axis indicates the ratio of Cd81 to Gapdh. Three independent tests were performed. CK, control; NC, a non-targeting siRNA used as a negative control. The siRNA-1 (CD81-human-498), siRNA-2 (CD81-human-168), and siRNA-3 (CD81-human-636) target distinct regions of the CD81 mRNA. ns: not significant. ****P < 0.0001.

Detection of apoptosis

The cells were incubated for 48 h at 37 °C/5% CO2. The AMKL cells were then washed with PBS. The cells were suspended in 1 ×binding buffer, stained with Annexin V-FITC and PI, and apoptosis was detected using a flow cytometer (the flow cytometry raw data were deposited in FlowRepository at FR-FCM-Z9ZQ.

Statistical analysis

Data were analyzed using GraphPad Prism. A paired t-test was performed to compare two samples. P-values <0.05 were considered statistically significant.

Figure 2 Screening for the virus MOI.

After 72 h of lentiviral infection, the infection efficiencies were shown at MOI of 0, 25, 50, 100, and 200. The top line shows the optical microscopic images of the UT-7 cells under different level of lentivirus. The middle line shows the fluorescence microscopic images of infected UT-7 cells. The bottom line shows the merges of the top and the middle images. Three independent tests were performed. MOI, multiplicity of infection.

Figure 3 Construction of stable CD81 knockdown cell line.

The horizontal axis indicates different group of UT-7 cells, and the vertical axis indicates the ratio of Cd81 to Gapdh. The CD81 expression of UT-7 cells with shCD81 was downregulated significantly compared with UT-7 CK or UT-7 NC cells. Three independent tests were performed. ****P < 0.0001.

Results

Construction of stable CD81 knockdown cell line

To screen the best target sequence, we transfected UT-7 cell line with three CD81 siRNA (siCD81). We compared the interfering efficiency of the control group (CK), siNC (a non-targeting siRNA used as a negative control), siRNA-1, siRNA-2, and siRNA-3. The expression of CD81 in UT-7 cells was detected by RT-PCR. As shown in Fig. 1, siRNA-2 demonstrated the best knockdown efficiency and its sequence was therefore selected for further research.

After 72 h of viral infection, the infection efficiency was greater than 80% when the multiplicity of infection (MOI) was 100 (Fig. 2). In this condition, cells with shCD81 significantly downregulated CD81 expression when compared with UT-7 cells (Fig. 3).

CD81 knockdown inhibits proliferation, inducts apoptosis, and decreases S phase cells of AMKL cell line UT-7

We investigated whether CD81 knockdown would inhibit the proliferation of AMKL cells. The AMKL cell line UT-7, including the CK, shRNA negative control (NC), and CD81-shRNA knockdown groups, was cultured and its proliferation capacity was assessed.

To further confirm the effects of CD81 expression on leukemic cell proliferation, UT-7 cells were transfected with shRNAs against CD81 and the knockdown efficiencies were validated by RT-PCR. As shown in Fig. 4, cells with reduced CD81 expression exhibited lower proliferation rates in CCK-8 assays.

Figure 4 The cell viability of UT-7 cells with shCD81 was downregulated.

(A) CCK8 analysis in UT-7 CK, NC, and CD81-shRNA knockdown groups. The horizontal axis indicates different group of UT-7 cells; the vertical axis indicates the proliferation percentage of the control. Three independent tests were performed. Cell viability was downregulated following CD81 was knockdown. (B) The optical microscopic image of the UT-7 cells. ****P < 0.0001. NC, shRNA negative control.

Figure 5 shows that the knockdown of CD81 in UT-7 was associated with an induction of apoptosis detected by increased Annexin V. The P value was less than 0.0001 when comparing the shCD81 and NC groups.

Figure 5 The rate of apoptosis was upregulated in UT-7 cells with shCD81.

(A) Bar chart of the rate of apoptosis in the UT-7 CK, NC, and CD81-shRNA knockdown groups. The horizontal axis indicates different groups of UT-7 cells; the vertical axis indicates the rate of apoptosis. shCD81 significantly upregulated the rate of apoptosis in the UT-7 cell line. (B) The cytometry analysis of the UT-7 CK, NC, and shCD81 groups. Three independent tests were performed. ns: not significant. ****P < 0.0001.

Figure 6 shows that CD81 knockdown inhibited cell proliferation with decreased S phase cells. Inversely, the proportions of G1 and G2 phase cells were upregulated in shCD81 AMKL cells.

Figure 6 The cell cycle analysis of UT-7 cell line with shCD81.

Bar chart of percentage of the S phase in the UT-7 CK, NC, and CD81-shRNA knockdown groups. The horizontal axis indicates different group of UT-7 cells; the vertical axis indicates the percentage of cells in S phase. ***P < 0.001.

Collectively, knockdown of CD81 significantly suppressed proliferation rate of AMKL cells.

Discussion

Our experiments demonstrated that CD81 knockdown can inhibit the proliferation of the AMKL cell line UT-7. CD81 knockdown promotes the apoptosis of AMKL cells. CCK-8 tests showed that AMKL cells showed decreased vitality following CD81 knockdown. The S phase cells were declined in shCD81 cells.

CD81 is a cell surface protein that plays important roles in cancer. Abu-Saleh et al. and colleagues (2023) concluded that CD81 knockout inhibits metastasis in vivo and vitro in breast cancer. Quagliano et al. (2020) found that CD81 knockout induces chemosensitivity in ALL. Other studies have shown that CD81 is an adverse prognostic marker in AML and multiple myeloma (MM) (Boyer et al., 2016; Paiva et al., 2012). In a cohort of 230 MM patients, CD81 was detected in 45% of them. In a cohort of 134 AML patients, CD81 was detected in 69% of patients. The FAB type included M0, M1, M2, M4, M5, and M6, but not M7 (AMKL). Therefore, the expression of CD81 in AMKL (AML-M7) may warrant further research using a larger cohort.

Vences-Catalán et al. and colleagues (2019) revealed that CD81 was upregulated in B cell lymphoma, and targeting CD81 with 5A6 (the anti-CD81 mAb) inhibited lymphoma in vivo and in vitro (Küppers, 2019). More importantly, they found 5A6 was as effective as rituximab (the anti-CD20 mAb), a well-established medicine. Interestingly, few side effects were observed in this study, even though CD81 is widely expressed. This may be related to the significantly higher expression of CD81 in malignant cells compared to normal cells. We speculate that the mechanism by which knockdown CD81 inhibits AMKL proliferation may be similar to the mechanism by which it participates in lymphoma (Vences-Catalán et al., 2019; Küppers, 2019). That is, knocking down CD81 may be involved in the direct killing and antibody-dependent cell cytotoxicity (ADCC) of AMKL cells. It may also activate caspase-3 and its downstream PARP targets. However, further experiments are needed for confirmation of this hypothesis.

Our previous work showed that CD81 is overexpressed in AMKL cells. However, there have been few studies on the correlation between CD81 and leukemia, especially regarding whether CD81 can be used as a target for AML. The prognosis of AMKL is extremely poor, and there is an urgent need to find better treatment options. Our experiment offers a possibility that CD81 may be one of the targets of AMKL. Although CD81 is widely expressed in cells, previous studies have shown that targeting CD81 results in minimal side effects. We speculate that when CD81 is combined with a specific marker of AMKL tumor cells, such as CD41 or CD61, as a target of a therapy such as CAR-T, it may be possible to specifically eliminate AMKL tumor cells.

Conclusion

In summary, CD81 may be a potential target of AMKL. These results provide new insights into the role CD81 plays in AMKL. The limitation of this experiment is that we have not yet investigated how CD81 is involved in the pathway of inhibiting AMKL. Since CD81 is located on the cell membrane of AMKL and there are readily available antibodies to CD81, future research should focus on whether CD81-targeting antibodies or even drugs can inhibit AMKL in the later stages. Mouse model experiments should be used to test whether CD81 has significant effects on normal cells.

Supplemental Information

Supplemental Information 1 The sequences of CD81-siRNAs and primers

Supplemental Information 2 MIQE checklist

Additional Information and Declarations

Competing Interests

Author Contributions

Data Availability

The authors declare there are no competing interests.

Narun Su conceived and designed the experiments, analyzed the data, prepared figures and/or tables, authored or reviewed drafts of the article, and approved the final draft.

Xiaohao Hu performed the experiments, prepared figures and/or tables, and approved the final draft.

The following information was supplied regarding data availability:

The raw data are available in the Supplementary File.

The flow cytometry data is available at FlowRepository, ID: FR-FCM-Z9ZQ.

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
