# Peer review of "CD81 inhibits the proliferation of acute megakaryoblastic leukemia cells"

_PeerJ, doi:10.7717/peerj.20286_

## Round 0.1 · original submission · Major Revisions

· Academic Editor

Major Revisions

**Language Note:** The review process has identified that the English language must be improved. PeerJ can provide language editing services - please contact us at [email protected] for pricing (be sure to provide your manuscript number and title). Alternatively, you should make your own arrangements to improve the language quality and provide details in your response letter. – PeerJ Staff

Reviewer 1 ·

Basic reporting

The authors present an interesting study on the role of CD81 in the proliferation and cell cycle regulation of AMKL (acute megakaryoblastic leukemia) cells.

While the work emphasizes the significance of CD81 in AMKL, the manuscript would benefit from a clearer articulation of how this research advances existing knowledge regarding CD81's role in AML and AMKL. Specifically, the novelty and contribution relative to previously published studies should be more explicitly stated.

Additionally, there is no mention of the supplementary figures. All figures and tables should be referenced properly.

Experimental design

Only one cell line was studied. Why?

While the experiments are planned to answer the question about the role of CD81 in AMKL, there needs to be more clarification of the methodology. The number of times and experiment was repeated is not clear. The figures definitely need improvement.

Validity of the findings

The results are well supported by the data. An additional literature review might be presented to support the findings.

Additional comments

Specific Comments

1. Grammar and Scientific Writing:
The manuscript would benefit from comprehensive editing for grammar, clarity, and scientific tone to improve overall readability.

2. References (e.g., Line 33): Ensure all references are properly cited. For instance, in line 33, the reference is unclear, and the sentence should specify that the increased expression was observed in T cells.

3. Line 51 – Clarify “LV-shCD81”: Expand the description to explain that LV-shCD81 refers to a lentiviral vector encoding a short hairpin RNA (shRNA) specifically designed to knock down CD81 expression.

4. Line 55 – Puromycin Selection: Indicate the duration of puromycin selection and the concentration used, as this is important for reproducibility.

5. Line 71 – Fixation Protocol: Clarify whether ethanol was added slowly to pre-washed cells (preferably while vortexing), and include the final concentrations of both Propidium Iodide (PI) and RNase A.

6. Lines 77–79 – Apoptosis Analysis: The methods mention only PI staining, which detects late apoptosis or necrosis, but Figure 2 shows both PI and Annexin V-FITC. Please revise the text to reflect that Annexin V-FITC staining was also performed.

7. Lines 81–82 – Statistical Analysis: Rephrase to improve clarity:
“A paired t-test was performed to compare two samples. P-values < 0.05 were considered statistically significant.”

8. Line 85 – Stable vs. Transient Knockdown: If siRNAs were used, they generally produce transient knockdown, not stable cell lines. Please clarify this discrepancy. In the methods, there is mention of shCD81 and siRNA, while the results only mention siRNA. Clarify.

9. Line 86 – Define siNC: Clearly define siNC as a non-targeting siRNA used as a negative control.

10. siRNA Design – Target Regions: The siRNAs used (CD81-human-498, -168, -636) should be described in the methods. State that they target distinct regions of the CD81 mRNA, which helps rule out off-target effects.

11. Figure 2 – Panel Labels: The figure panels are unclear. If they represent different time points, they should be labeled accordingly in both the figure and the legend.

12. Clarity of Expression – Line 90: Rephrase for clarity: “After 72 hours of viral infection, the infection efficiency was greater than 80% when the multiplicity of infection (MOI) was 100 (Figure 2).”

13. Figure 3 – Statistical Notation: Instead of using symbols like **# and *** to indicate significance, use standard significance bars drawn between groups (e.g., CK vs. siCD81), along with consistent symbols to denote P-value.

14. Line 95–96 – Describe Experimental Design Correctly: You are not “treating” cells but using previously generated knockdown cell lines. A clearer phrasing would be:
“The AMKL cell line UT-7, including the control (CK), siRNA negative control (NC), and CD81-siRNA-2 knockdown groups, was cultured, and their proliferation capacity was assessed.”

15. Lines 94–106 – Broader Focus: This section discusses cell proliferation, apoptosis, and the cell cycle, while the title refers only to proliferation. Either revise the title to reflect all outcomes or focus the text accordingly.

16. CD81 and Prognosis: There is no mention that CD81 is already associated with adverse prognosis in AMKL. Please include relevant references

Additional Points

17. Figure Legends:
The legends are incomplete and should be revised to fully describe the experimental conditions, statistical comparisons, and what each panel shows.

18. When there are multiple parts in a figure, label them as A, B, …

Reviewer 2 ·

Basic reporting

1. What is the prevalence of AMKL among the Chinese population?

2. What is the overall survival of patients diagnosed with AMKL?

3. Define the methodology adopted for the diagnosis of AMKL.

4. What are the current therapeutic options available for the treatment of AMKL?

5. Is there any mechanism of CD81 and its role in the pathogenesis of AMKL?.

Experimental design

1. Specify the RNA concentration used for cDNA synthesis.

2. Indicate the statistical software used for data analysis.

Validity of the findings

1. Elaborate the discussion section with detailed explanations of relevant studies and their conclusions regarding CD81 and its role in leukemia.

2. Expand and improve the results section. Present the findings under more detailed subheadings for clarity and completeness.

---

## Round 0.2 · accepted · Accept

· Academic Editor

Accept

All issues pointed out by the reviewers were addressed and the revised manuscript is acceptable now.

Reviewer 2 ·

Basic reporting

It seems fine. All comments were properly incorporated into this revised manuscript.

Experimental design

-

Validity of the findings

-